# An Innovative Approach for Removing Stripe Noise in Infrared Images

**DOI:** 10.3390/s23156786

**Published:** 2023-07-29

**Authors:** Xiaohang Zhao, Mingxuan Li, Ting Nie, Chengshan Han, Liang Huang

**Affiliations:** 1Changchun Institute of Optics, Fine Mechanics and Physics, Chinese Academy of Sciences, Changchun 130033, China; zhaoxiaohang21@mails.ucas.ac.cn (X.Z.); limingxuan17@mails.ucas.ac.cn (M.L.); nieting@ciomp.ac.cn (T.N.); hanchengshan@ciomp.ac.cn (C.H.); 2University of Chinese Academy of Sciences, Beijing 100049, China

**Keywords:** infrared images, stripe noises, multi-level image decomposition method, multi-sparse constraint representation (MSCR), Alternating Direction Method of Multipliers (ADMM)

## Abstract

The non-uniformity of infrared detectors’ readout circuits can lead to stripe noise in infrared images, which affects their effective information and poses challenges for subsequent applications. Traditional denoising algorithms have limited effectiveness in maintaining effective information. This paper proposes a multi-level image decomposition method based on an improved LatLRR (MIDILatLRR). By utilizing the global low-rank structural characteristics of stripe noise, the noise and smooth information are decomposed into low-rank part images, and texture information is adaptively decomposed into several salient part images, thereby better preserving texture edge information in the image. Sparse terms are constructed according to the smoothness of the effective information in the final low-rank part of the image and the sparsity of the stripe noise direction. The modeling of stripe noise is achieved using multi-sparse constraint representation (MSCR), and the Alternating Direction Method of Multipliers (ADMM) is used for calculation. Extensive experiments demonstrated the proposed algorithm’s effectiveness and compared it with state-of-the-art algorithms in subjective judgments and objective indicators. The experimental results fully demonstrate the proposed algorithm’s superiority and efficacy.

## 1. Introduction

The non-uniformity of infrared detectors often appears as stripe noise in images, which directly affects image quality and even impedes subsequent image processing tasks, such as image classification, target detection, and recognition [1,2,3]. Therefore, removing stripe noise while retaining fine image details is particularly important. This paper aims to separate the stripe noise component from the target infrared image and preserve the delicate texture details of the effective information in the image.

Currently, methods for removing stripe noise can be roughly divided into four categories: filter-based, statistics-based, optimization-based, and neural-network-based. The method proposed in this paper belongs to the optimization-based approach.

Q.U.A.R.M.B.Y. introduced the filter-based method for removing stripe noise in 1987 [4]. Subsequently, various filtering methods were developed, such as Fourier filters [5], wavelet analysis [6], and wavelet-Fourier combined filters. Boyang Chen proposed an adaptive wavelet filter to quickly and accurately learn appropriate wavelet filter coefficients [7]. Ende Wang proposed a stripe removal algorithm based on wavelet decomposition and gradient equalization [8]. However, no filter can perfectly accommodate all frequencies of stripe noise, and some image components containing information will inevitably be filtered out, causing the resulting clean image to lose some information or produce “artifacts”.

Statistics-based methods for removing stripe noise are often widely used in engineering practices. This method statistically assumes that all infrared detectors’ response expectations and variance are consistent. For example, Carfantan proposed a new self-calibration and stripe removal technique for push-broom satellite imaging systems that does not require specific training data but instead assumes linear responses based on statistical estimates of the gain of each detector in the observed image [9]. The algorithm of the statistics-based stripe noise removal method is relatively simple and highly applicable, but its effectiveness in removing stripe noise in infrared images is poor.

Optimization-based stripe noise removal methods are currently the most effective method in all of the research. Leonid I. Rudin initially proposed a numerical algorithm for removing image noise based on constrained optimization [10]. Subsequently, Fa Rsiu S used L1 norm minimization and robust regularization to deal with different data and noise models [11]. Yong Chen proposed a new stripe removal method that fully considers the inherent characteristics of stripe noise and image features [12]. According to the researchers’ in-depth exploration, stripe noise has global sparsity and gradient sparsity along the stripe direction. Li Mingxuan used the L1 norm to sparsely represent stripe noise, forming a regularization term in the energy function, and used a sparse representation of clean images across stripes as fidelity terms to separate and minimize stripe noise, combined with methods such as the wavelet transform, resulting in a good denoising ability [13,14].

The initial application of neural network methods in infrared image non-uniformity correction was proposed by Scribner in 1991 [15]. Subsequently, Vera proposed an adaptive scene-based non-uniformity correction technique [16]. Rossi proposed using the image obtained by filtering with a bilateral filter as the expected image [17]. K Xu et al. proposed a deep multi-scale dense connection convolutional neural network (DMD-CNN) [18]. However, the neural-network-based stripe noise removal method is highly required to select implicit target images in the algorithm and design loss functions in the correction layer.

Noise removal is a prerequisite for contrast or detail enhancement in infrared images [19]. Existing stripe noise removal methods for infrared images rarely consider the global low-rank characteristics of the stripe noise itself. Therefore, many stripe noise removal algorithms for infrared images currently either have an insufficient denoising ability, resulting in processed images still containing stripe noise, or they are over-denoised, resulting in the loss of effective texture information. Based on previous research, this paper aims to remove stripe noise while ensuring that the effective information in non-stripe noise areas of the image is not affected, fully utilizing the structural information of stripe noise and proposing a more practical algorithm for removing stripe noise in infrared images.

The latent low-rank representation (LatLRR) [20] is usually used for clustering analysis tasks. Some scholars have proposed better subspace recovery methods based on LatLRR [21]. The authors of [20] also mentioned that LatLRR can extract salient and low-rank components from input data. Based on the detail extraction capability of LatLRR, several scholars have proposed various image decomposition and fusion strategies. Liu et al. proposed a fusion method based on multi-decomposition LatLRR. They combined it with a dual-simplified pulse-coupled neural network (D-SPCNN) to improve the fusion of detail layers [22]. Li et al. proposed a new infrared and visible image fusion method based on saliency detection and LatLRR-FPDE [23].

In this paper, based on the globally low-rank structure of stripe noise, MIDILatLRR was proposed, which adapts the selection of decomposition levels to extract different levels of delicate texture information from the salient components at each level, while the low-rank components at the last level focus on extracting all the stripe noise and low-rank smooth parts in the image. Based on this, an optimal model called the MSCR of stripe noise was constructed according to the sparse difference in directionality and smoothness between the stripe noise and effective information in the last-level low-rank part of the image and used the L1 norm to constrain sparse terms. The alternating direction multiplier method (ADMM) was used to solve this model [24,25].

The main contributions are summarized as follows:(a)We experimentally demonstrate and analyze the effectiveness and underlying mechanisms of using LatLRR decomposition to extract the low-rank component containing stripe noise in noisy infrared images while preserving the rich texture information in the salient component. Based on this, MIDILatLRR was proposed to fully extract the effective texture information in infrared images, separating the stripe noise from useful information. This approach only requires denoising the final low-rank part image while maintaining the texture details in salient part images.(b)A denoising model was proposed based on MSCR to address the difference between the effective smooth and stripe noise parts in the final low-rank part image obtained through MIDILatLRR. The MSCR model takes advantage of the sparsity of stripe noise, the smoothness of effective information present in the final low-rank component, and the sparse difference in gradients in different directions to impose the sparsity constraint and extract the noise component.(c)In solving MIDILatLRR, an adaptive level decomposition was established based on the richness of the detail features in different input noisy images, making it an adaptive cut-off in the decomposition, and the chosen level of decomposition is convergent. In the MSCR model, the L1 norm was used to constrain the directional gradient sparsity of stripe noise and the edge sparsity of low-rank smooth information, which generates a non-convex optimization model. This model is solved using the ADMM.

The remainder of the paper is organized as follows: Section 2 presents the proposed method’s theoretical principles. Section 3 describes the main content flow of the algorithm. Section 4 selects noisy images and several state-of-the-art algorithms to experimentally validate the performance of the proposed algorithm through ablation experiments and comparative experiments. Finally, Section 5 provides the conclusion.

## 2. Preliminaries

In this chapter, the core mechanism of MIDILatLRR was experimentally and mathematically validated by demonstrating and analyzing the ability of LatLRR to separate stripe noise and complicated texture information in noisy infrared images, laying the foundation for the method in the next chapter.

### 2.1. Latent Low-Rank Representation

In 2010, LIU et al. [26] proposed the LRR subspace segmentation theory, in which the input data matrix itself is chosen as the dictionary. However, this method brings a series of issues, such as the dictionary needing to contain sufficient data vectors sampled from subspaces, and the input data cannot be corrupted. Otherwise, the subspace segmentation may fail.

Therefore, in 2011, the authors [20] proposed the LatLRR theory, which can decompose the original data into low-rank and significant structural components. The formulation of the LatLRR problem is as follows:(1)Z∗ +L∗ +λE1 Z,L,E min s.t. X=XZ+LX+E

In the equation, λ > 0 is a balance coefficient, ${\left\|\;\right\|}_{1}$ represents the L1 norm, and ${\left\|\;\right\|}_{*}$ represents the nuclear norm, the sum of the matrix singular values. X represents the observed data matrix, Z is a matrix of low-rank coefficients, L is a projection matrix named a salient coefficients matrix, and ***E*** is a sparse, noisy matrix. Equation (1) is solved by the inexact Augmented Lagrangian Multiplier [20], and then the XZ low-rank component and LX salient component can be obtained based on the equation.

The LatLRR algorithm, which possesses an unsupervised feature extraction capability and non-dimensionality reduction property, was initially discovered by its author, Liu. The salient component of LatLRR holds discriminative power for essential data information in a matrix. Although this non-dimensionality reduction property is not conducive to feature information extraction in application domains, it has a relatively wide range of applications in the field of image processing.

The original authors did not provide much explanation for the feature extraction capability of the LatLRR algorithm itself. Yaming Wang et al. [27] have explained and conducted further studies on the feature extraction characteristics of LatLRR. The feature extraction capability of LatLRR finds wide use in image fusion. Hui Li et al. [28] proposed a multi-level decomposition strategy (MDLatLRR) based on LatLRR that extracts basic parts and significant features at different levels of the image, improving the performance of image fusion and fully utilizing the feature extraction capacity of LatLRR.

In addition to feature extraction and image fusion, this study reveals that LatLRR can potentially remove stripe noise in infrared images. The stripe noise caused by the non-uniformity of the readout circuitry in infrared images demonstrates globally low-rank structure characteristics. Through the LatLRR decomposition, stripe noise can be extracted into low-rank components.

### 2.2. Experimental Verification and Analysis

In Equation (1), if an infrared image containing stripe noise is input into ***X***, it will be discovered through experiments that the stripe noise can be decomposed into an ***XZ*** low-rank component by LatLRR. The ***XZ*** part primarily contains the low-frequency component of the image, which reflects the main body of the original image after LatLRR decomposition. Meanwhile, the high-frequency information component of the image, reflecting the local detailed texture part of the original image, can be primarily extracted by the ***LX*** part. Since the stripe noise in the infrared image has continuity in the vertical direction and low-frequency characteristics, it belongs to the global structure. Therefore, the ***XZ*** low-rank component can always extract the stripe noise in the noisy infrared image after LatLRR decomposition.

From Figure 1, it is evident that the main information of the image and the stripe noise are contained in the decomposed low-rank component, while the salient component primarily reflects the local information of the image. Furthermore, the sparse noise matrix ***E*** exhibits a significant sparsity level and is typically used for separating blind elements and salt-and-pepper noise in infrared images. However, the current infrared images have few elements in this part, and the ***λ*** coefficient selection further restricts the extraction of the sparse noise component ***E*** for image elements. As a result, the extracted image information from the ***E*** component displays high sparsity. In these experiments, the L1 norm of each column of the sparse noise component ***E*** compared to the original image with a λ value of 1 was plotted, as depicted in Figure 1.

From Figure 1d,e, it can be observed that the impact of the sparse noise matrix on the original image is minimal. Therefore, to facilitate the verification and analysis of the stripe noise within the low-rank component, the LatLRR problem was simplified to the following equation:(2)Z∗+L∗Z,L,Emins.t.X=XZ+LX

This equation only decomposes the input data into the low-rank component, ***XZ***, and the salient component, ***LX***, making it easier to analyze the principle of LatLRR for feature extraction. Zhang et al. analyzed this equation [29] and derived a closed-form solution for noiseless LatLRR. In the following theorem, we reaffirm the main result of noiseless LatLRR.
(3)$$X=U_{X}\Sigma_{X}V_{X}^{T},Z=V_{X}W_{Z}V_{X}^{T},\;L=U_{X}(1-W_{Z})U_{X}^{T}$$
where $U_{X}$ and $V_{X}^{T}$ are the left and right singular matrices of input matrix ***X***, respectively, while $\Sigma_{X}$ is the singular value matrix of ***X***, and $W_{Z}$ is any block diagonal matrix satisfying: 1. If ${\left[\Sigma_{X}\right]}_{ii}\neq {\left[\Sigma_{X}\right]}_{jj}$ then ${\left[W_{Z}\right]}_{ij}=0;$ 2, both $W_{Z}$ and $\left(1-W_{Z}\right)$ are positive semi-definite [29].

Notice that in practice, the singular values of ***X*** are usually distinct; therefore, $W_{Z}$ becomes a diagonal matrix $diag\{z_{1},z_{1},...,z_{r}\}with0\leq z_{i}\leq 1$ for all i.

Then, ***XZ*** and ***LX*** can be viewed as obtained through formula manipulation, which can be expressed as follows:(4)$$XZ=U_{X}\Sigma_{X}W_{Z}V_{X}^{T}\;=U_{X}\left[\begin{array}{ccc} z_{1}\sigma_{Q1} & \cdots & 0\\ \vdots & \ddots & \vdots \\ 0 & \cdots & z_{r}\sigma_{Qr} \end{array}\right]V_{X}^{T}$$
(5)$$LX=U_{X}(1-W_{Z})\Sigma_{X}V_{X}^{T}\;=U_{X}\left[\begin{array}{ccc} l_{1}\sigma_{X1} & \cdots & 0\\ \vdots & \ddots & \vdots \\ 0 & \cdots & l_{r}\sigma_{Xr} \end{array}\right]V_{X}^{T}$$
here $l_{i}=1-z_{i}$, and $\sigma_{Qi}$ is i th largest singular value of ***X***.

In this, we can observe that the fundamental principle underlying the separation of the low-rank and salient parts in the decomposition Formula (3) is the weighted processing of the singular values of the original image based on their relative magnitudes. Mappings of $l_{i}$ and $z_{i}$ with different weights for singular values of diverse magnitudes are utilized; meanwhile, $W_{L}=1-W_{Z}$ suppresses the weight component associated with a larger $\sigma_{Xi}$. Afterward, the weight ratios of $z_{i}$ and $l_{i}$ separated from a noisy infrared image are illustrated in Figure 2.

Through an examination of Figure 2, it can be observed that the weight ratio of the salient component, $l_{i}$, is relatively small in the first several singular values. Meanwhile, detailed texture information of the image is contained in the singular values with smaller numerical values and occurs later in the order [30]. Subsequently, experiments are conducted to analyze which singular values in the LatLRR decomposed image matrix represent the stripe noise information captured in the original image. It is worth noting that the left and right singular matrices utilized for the singular value decomposition of the stripe noise matrix at this stage are the same as those used for the singular value decomposition of the original image.

By performing a singular value decomposition of the stripe noise presented in the Figure 3, it can be concluded that the noise in images exhibiting stripe noise primarily manifests in the relatively large singular values of $\Sigma_{X}$. It can be inferred that the stripe noise present in infrared images comprises one of the main constituents of the image and exhibits relatively low-frequency characteristics. Additionally, through a comparison of Figure 2 and Figure 3, it is observed that the salient components extracted by LatLRR fail to capture the primary information contained in the large singular values corresponding to the stripe noise. Moreover, this Figure 3 illuminates that LatLRR decomposes the image using varying weights for singular values of distinct magnitudes.

In the forthcoming section, a qualitative analysis will be conducted by contrasting the low-rank and salient components presented in Figure 4. As the salient component matrix comprises negative elements, a normalization technique is implemented to scale the values to the range from 0 to 1 for visualization.

The salient component of Figure 4, after normalization, demonstrates excellent capabilities for extracting details in both fine texture information and vertical texture information in the background while completely separating the stripe noise. Although the normalization results successfully extract image detail texture information, this global technique inevitably causes distortion in part of the mapped image. Additionally, extracting salient components from a single LatLRR decomposition cannot extract all of the image’s detailed information. Therefore, MIDILatLRR was proposed by further improving the algorithm based on the detail extraction and stripe noise separation abilities of LatLRR. MIDILatLRR can completely extract the fine texture details of the image while concentrating the stripe noise in the final low-rank part image, making it more conducive to subsequent denoising algorithm processing. The main flowchart of the proposed algorithm is illustrated by Figure 5 as follows:

## 3. Primary Algorithms

In the previous chapter, the ability of LatLRR to separate stripe noise and effective texture information was validated. Based on this characteristic, a more effective multi-level graphic decomposition method, MIDILatLRR, is constructed in this chapter. After decomposition, an analysis of sparse terms is conducted in the final low-rank part image. An MSCR model for removing stripe noise is established and solved.

### 3.1. Multi-Level Image Decomposition Method Based on Improved LATLRR

As previously mentioned, a single LatLRR decomposition has limited capability in extracting salient information, and direct normalization of the salient component leads to image distortion. Hence, MIDILatLRR has been designed to fully extract the detailed information of the image and concentrate the stripe noise in the low-rank component image of the final level. This approach is more conducive to protecting useful information and separating stripe noise in noisy infrared images. Moreover, it fully utilizes the property that stripe noise exists only in the low-rank component after LatLRR decomposition. The algorithm formula is presented below:(6)$$D_{i}=K_{i}\left(Ep\left({{LX}_{i}}^{+}\right)\right)$$
$$\;s.t.min\left(X_{i}-D_{i}-E_{i}\right)=0$$
(7)$$X_{i+1}=X_{i}-D_{i}$$

The extraction of positive matrix elements from the salient component matrix ***LX*** is represented by $Ep\left({{LX}_{i}}^{+}\right)$, which is followed by a linear mapping to the [0-1] interval. The mapping scale factor for $Ep\left({{LX}_{i}}^{+}\right)$ is defined as the coefficient $K_{i}$, which ranges from 0 to 1. To ensure that the extracted images are not distorted, the value of $K_{i}$ is chosen to satisfy Formula (6)‘s condition while being maximized. Details are extracted from the image without distortion using this method, as illustrated in Figure 6. Repetitive multi-level image decomposition is performed by subtracting the extracted information from the original image using $X_{i+1}=X_{i}-D_{i}$, and they are using it as an input for the next level of LatLRR decomposition. As a result, the image details that are contained in the negative elements of the salient component matrix that were not extracted in previous levels are continuously extracted in subsequent levels. Hence, this method effectively extracts maximum information from noisy infrared images without distortion.

The positive elements of the salient component matrix ${LX}_{i}$ are extracted, and the distortionless linearly mapped matrix $D_{i}$, which represents the *i*-th level salient part image, is obtained. The $X_{i}-D_{i}$ part of the image where stripe noise is more concentrated is referred to as the *i*-th level low-rank part image, $V_{i}$. As shown in Figure 6, multi-level image decomposition is performed using MIDILatLRR. With increasing levels, texture detail information gradually reduces in the low-rank part, $V_{i}$, resulting in a smoother image. In $V_{4}$, smooth blocks and stripe noise represent most of the image information; further, MIDILatLRR image decomposition is not meaningful. Therefore, different image decomposition levels should be selected for images with different texture richness.

### 3.2. Establishment of Sparse Regularization Terms for MSCR

After image decomposition, the proposed MSCR denoising algorithm in this section only targets the final low-rank part image, avoiding the processing of complex texture information in salient part images.

In the final low-rank part image, the stripe noise was observed to have strong directional characteristics. In contrast, the non-noisy effective information component of the final low-rank part image presented relatively smooth characteristics and extensive block features. In light of this, the corresponding regularization terms for stripe noise removal were designed in the final low-rank part image by constraining the features of noise or effective information to represent the stripe noise based on the difference between stripe noise and effective information in the image. Figure 7 illustrates some characteristics of the stripe noise and effective information in the final low-rank part image. Here, the difference between stripe noise and smooth information can be observed.

#### 3.2.1. Smoothness of Effective Information

In this paper, the stripe noise matrix is represented as ***N***, and the final low-rank part image of the nth level is denoted as $V_{n}$. The effective information component of the final low-rank part image is represented by $V_{n}-N$. As shown in Figure 7b,d,e, there are noticeable differences in smoothness between the effective information and the stripe noise in the final low-rank part image, $V_{n}$. Moreover, the texture information has been fully extracted by salient part images of MIDILatLRR. Therefore, this image component appears relatively smooth, and it does not contain prominent texture information in the absence of stripe noise. Figure 7d,e demonstrates that the effective information component becomes sparse when convolved with the Laplace edge detection operator, which leads to a large difference in the L1 norm between the effective information component and the original image. The smoothness of the effective information component in the final low-rank part image, $V_{n}$, was used as a fidelity term and enforced sparsity using the L1 norm:(8)$$P_{1}\left(N\right)={\left\|d_{L}V_{n}-d_{L}N\right\|}_{1}$$

In this Laplace operator, $d_{L}$ was utilized for image edge detection, while ***N*** represented the stripe noise component present in both the $V_{n}$ image matrix and the original image. To calculate the sparsity of ***N***, the L1 norm was adopted since it is easier to implement than the L0 norm. 

Additionally, as Figure 7i shows, the effective information component in the final low-rank image, $V_{n}$, had a smaller proportion of L1 norms per row after being convolved with the edge detection operator. This finding indicated that the constraint term enforced by the L1 norm could ensure the overall smoothness of the effective information after denoising.

#### 3.2.2. The Sparsity and Directionality of Stripe Noise

As can be seen in Figure 7c, the stripe noise in the infrared image is present in a columnar form. The pixels in the regions without stripes have a value of zero, which makes the stripe noise component sparse and suitable for sparsity constraints. The L1 norm was utilized to represent this regularization term to avoid excessive denoising, as it has better non-convexity. The representation is shown below:(9)$$P_{2}\left(N\right)={\left\|N\right\|}_{1}$$

Furthermore, Figure 7j reveals that the L1 norm of each column in the noise component is comparatively smaller than that of the original image, indicating that a sparse constraint can be introduced.

Due to the strong directionality of stripe noise, the horizontal gradient of the final low-rank component image, $V_{n}$, predominantly originates from the stripe noise, as depicted in Figure 7f,g. In the stripe noise component image, clear vertical stripes can be observed in the gradient domain along the horizontal direction. Conversely, the effective information component exhibits a lower proportion of the original image, $V_{n}$, in the horizontal gradient domain and is relatively sparse along the horizontal gradient domain. Hence, the difference between the horizontal gradients of the final low-rank component image and the stripe noise component was employed as a fidelity term and applied to the L1 norm to enforce sparsity constraints, as shown below:(10)$$P_{3}\left(N\right)={\left\|d_{x}V_{n}-d_{x}N\right\|}_{1}$$

In addition, Figure 7i demonstrates that the proportion of the L1 norm of the horizontal gradient of the effective information component relative to that of the original image is relatively small. Thus, horizontal smoothing of the denoised image can be achieved using this regularization term.

### 3.3. Multi-Sparse Constraint Representation Model

As analyzed above, significant differences exist in structure and directionality between the stripe noise component and information component in infrared images. Combined with the three components above, $P_{1}\left(N\right)$, $P_{2}\left(N\right)$, $P_{3}\left(N\right)$, the final optimization model for stripe noise was obtained:(11)$$N=arg\underset{N}{\min}\lambda_{1}{\left\|d_{L}V_{n}-d_{L}N\right\|}_{1}+\lambda_{2}{\left\|N\right\|}_{1}+\lambda_{3}{\left\|d_{x}V_{n}-d_{x}N\right\|}_{1}$$

The equation incorporates three regularization parameters, $\lambda_{1}$, $\lambda_{2}$, and $\lambda_{3}$, which are utilized to balance each term. The minimum stripe noise component, N, that minimizes Equation (11) is solved for and extracted first. Finally, the denoised information components $V_{n}\prime$ can be estimated through the transform below:(12)$$V_{n}\prime =V_{n}-N$$

### 3.4. Solution Process

The denoising mathematical model has been established in the previous section. This section will use the adaptive determination method for MIDILatLRR and solve the MSCR model using the ADMM method.

#### 3.4.1. Adaptive Determination of the Decomposition Level for MIDILatLRR

The *i*-th level salient part image $D_{i}$, which contains the details and texture information of the image, was decomposed by the MIDILatLRR algorithm. To verify that most of the texture information in the image can be extracted by the first several levels of image decomposition, convergence experiments were conducted. The *i*-th level salient part image $D_{i}$ from the first n levels was summed up to obtain the image ${DS}_{n}$, which contains more texture detail information.
(13)$${DS}_{n}=\sum_{i=1}^{n}D_{i}$$

Convolutions with the Laplace edge detection operator [31] were separately performed on ${DS}_{n}$ and the original image. The ratio of L1 norms after convolution was used to represent the richness of the salient part images extracted from the first n levels. As all the salient part images do not contain stripe noise, the L1 norm of the convolved sum of the salient component images, ${DS}_{n}$, must be smaller than that of the original image. Theoretically, when reaching a certain level of decomposition, this ratio will be fixed around a certain value, fluctuating around this value but not approaching 1. The difference between this ratio and one mainly comes from the L1 norm difference caused by stripe noise in the original image after convolution with the operator.

As depicted in Figure 8, the richness of ${DS}_{n}$ has achieved approximately 0.65 of the original image after undergoing four to five decompositions with MIDILatLRR, and the subsequent increase is comparatively insignificant. The maximum richness of ${DS}_{n}$ is achieved at the seventh decomposition level. However, the texture information contained in the salient components is neutralized due to the relatively smooth information extracted in the later decomposition levels, which results in a minor decrease in the richness ratio of ${DS}_{n}$. Based on an extensive experimental analysis, it was decided to terminate MIDILatLRR decomposition once the richness increase dropped below 5%. In this case, the decomposition level of this image was established at four. Through extensive image experiments, it has been verified that upon reaching this termination condition, the difference in richness between ${DS}_{n}$ and the original image is mainly due to the absence of stripe noise. At the fourth decomposition level, the low-rank part image $V_{4}$ contains almost solely stripe noise, and further decomposition only marginally affects the detail extraction of the salient components. Besides the stripe noise, $V_{4}$ constitutes large-area blocky smooth information, which attests to the salient components’ sufficient extraction of image details in MIDILatLRR.

After denoising $V_{4}$, the subsequent stripe noise removal algorithm requires an image as the separation of the image detail texture and low-rank information has been accomplished. In this section, sufficient complex texture information has been extracted from the original image through MIDILATLRR, as demonstrated in Figure 9b, where numerous detailed features, such as the texture of the trousers and facial features, have been appropriately extracted by MIDILATLRR. Solely the final low-rank part image $V_{4}$ necessitates denoising, thus avoiding any loss to the pre-extracted salient component ${DS}_{4}$.

#### 3.4.2. ADMM Optimization for MSCR

The most direct approach for solving optimization problems involves a convergence matrix, whereby some Equation uses second-order differentiation. However, for regularization models like Equation (7), which are based on L1 norm regularization, the regularization term is not continuously differentiable and therefore poses difficulties for differentiation. The ADMM algorithm is currently widely used in machine learning and other fields. Its essence is to optimize the unconstrained part using the block coordinate descent method. It is an effective method for solving the L1 norm regularization term. Therefore, this method is used to solve Equation (11). The solution process is shown as follows:

To address the three regularization terms, three auxiliary variables are introduced: $H=d_{l}V_{n}-d_{l}N$*,*
R=N, and $M=d_{x}V_{n}-d_{x}N$, so that the minimization of Equation (11) is equivalent to minimizing the following:(14)argminN,G,T,U⁡λ1H1+λ2R1+λ3M1s.t.H=dyN,R=N,M=dxVn−dxN

The convex optimization problem in the form of Equation (13) can be further transformed into an augmented Lagrangian function, i.e.,
(15)$$\arg\underset{N,G,T,U}{\min}\lambda_{1}{\left\|H\right\|}_{1}+\lambda_{2}{\left\|TR\right\|}_{1}+\lambda_{3}{\left\|M\right\|}_{1}+m_{1}^{T}\left(d_{L}V_{n}-d_{L}N-H\right)+m_{2}^{T}\left(N-R\right)+m_{3}^{T}\left(d_{x}V_{n}-d_{x}N-M\right)\;+\frac{\rho_{1}}{2}{\left\|d_{L}V_{n}-d_{L}N-H\right\|}_{2}^{2}+\frac{\rho_{2}}{2}{\left\|N-R\right\|}_{2}^{2}+\frac{\rho_{3}}{2}{\left\|d_{x}V_{n}-d_{x}N-M\right\|}_{2}^{2}\;$$

In this case, $m_{1},m_{2},m_{3}$ are the Lagrange multipliers for each constraint term while $\rho_{1},\rho_{2},\rho_{3}$ are the penalty term parameters. At this point, Equation (15) can be transformed into four different sub-items to be iteratively solved:a.***H*** problem
(16)$$H=arg\underset{G}{min}\lambda_{1}{\left\|H\right\|}_{1}+m_{1}^{T}\left(d_{L}V_{n}-d_{L}N-H\right)+\frac{\rho_{1}}{2}{\left\|d_{L}V_{n}-d_{L}N-H\right\|}_{2}^{2}$$

According to Formula (11) in Reference [32], for solving for X such that the following Formula is minimized:(17)$$arg\underset{X}{min}{\left\|X-B\right\|}_{2}^{2}+2\lambda {\left\|X\right\|}_{1}$$

It can be derived directly that
(18)$$X=soft\left(B,\lambda \right)=sign(B)max(\left|B\right|-\lambda ,0)$$

Therefore, Equation (16) can be converted into the following:(19)$$H=arg\underset{H}{min}\lambda_{1}{\left\|H\right\|}_{1}+\frac{\rho_{1}}{2}{\left\|d_{L}V_{n}-d_{L}N-H+\frac{m_{1}}{\rho_{1}}\right\|}_{2}^{2}$$

Following the solution principle of Formula (17), it can be solved by the following:(20)$$H^{k+1}=soft\left(d_{L}V_{n}{-d}_{L}N^{k}+\frac{m_{1}^{k}}{\rho_{1}},\frac{\lambda_{1}}{\rho_{1}}\right)$$
where *k* represents the number of iterations.


b.***R*** problem.

(21)
$$R=arg\underset{R}{min}\lambda_{2}{\left\|T\right\|}_{1}+m_{2}^{T}\left(N-R\right)+\frac{\rho_{2}}{2}{\left\|N-R\right\|}_{2}^{2}$$



Likewise, for the G problem, the solution is obtained as follows:(22)$$R^{k+1}=soft\left(N^{k}+\frac{m_{2}^{k}}{\rho_{2}},\frac{\lambda_{2}}{\rho_{2}}\right)$$


c.***M*** problem.

(23)
$$M=arg\underset{U}{min}\lambda_{3}{\left\|U\right\|}_{1}+m_{3}^{T}\left(d_{x}V_{n}-d_{x}N-M\right)+\frac{\rho_{3}}{2}{\left\|d_{x}V_{n}-d_{x}N-M\right\|}_{2}^{2}$$



The solution is obtained as follows:(24)$$M^{k+1}=soft\left(d_{x}V_{n}{-d}_{x}N^{k}+\frac{m_{3}^{k}}{\rho_{3}},\frac{\lambda_{3}}{\rho_{3}}\right)$$


d.The ***N*** problem.

(25)
$$N=arg\underset{N}{min}m_{1}^{T}\left(d_{L}V_{n}-d_{L}N-H\right)+m_{2}^{T}\left(N-R\right)+m_{3}^{T}\left(d_{x}V_{n}-d_{x}N-M\right)\;\;\;\;\;\;\;\;\;\;\;+\frac{\rho_{1}}{2}{\left\|d_{L}V_{n}-d_{L}N-H\right\|}_{2}^{2}+\frac{\rho_{2}}{2}{\left\|N-R\right\|}_{2}^{2}+\frac{\rho_{3}}{2}{\left\|d_{x}V_{n}-d_{x}N-M\right\|}_{2}^{2}$$



It can be simplified as follows:(26)$$N=arg\underset{N}{min}\left\{\frac{\rho_{1}}{2}{\left\|d_{L}V_{n}-d_{L}N-H+\frac{m_{1}}{\rho_{1}}\right\|}_{2}^{2}+\frac{\rho_{2}}{2}{\left\|N-R+\frac{m_{2}}{\rho_{2}}\right\|}_{2}^{2}+\frac{\rho_{3}}{2}{\left\|d_{x}V_{n}-d_{x}N-M+\frac{m_{3}}{\rho_{3}}\right\|}_{2}^{2}\right\}$$

This is a quadratic optimization with differentiability. It is equivalent to solving the following linear system. Through the direct derivation of Formula (26):(27)$$\rho_{1}{d_{L}}^{T}⨂d_{L}⨂N^{k+1}+\rho_{2}N^{k+1}+\rho_{3}{d_{x}}^{T}⨂d_{x}⨂N^{k+1}\;\;\;\;\;\;\;\;\;\;\;\;\;\;\;\;\;\;\;\;\;\;\;\;\;\;\;\;=\rho_{1}{d_{L}}^{T}⨂(d_{L}V_{n}-H^{k+1}+\frac{m_{1}}{\rho_{1}})+\rho_{2}\left(R^{k+1}-\frac{m_{2}}{\rho_{2}}\right)+\rho_{3}{d_{x}}^{T}⨂(d_{x}V_{n}-M^{k+1}+\frac{m_{3}}{\rho_{3}})$$
where ⨂ denotes convolution. It is not easy to solve a formula involving convolution. This paper introduces the Fourier transform to convert the convolution in the time domain into a multiplication in the frequency domain:(28)$$\left[\rho_{1}F\left({d_{L}}^{T}\right).*F\left(d_{L}\right)+\rho_{2}+\rho_{3}F\left({d_{x}}^{T}\right).*F\left(d_{x}\right)\right].*F\left(N^{k+1}\right)\;\;\;\;\;\;\;\;\;\;\;\;\;\;\;=\rho_{1}F\left({d_{L}}^{T}\right).*F(d_{L}V_{n}-H^{k+1}+\frac{m_{1}}{\rho_{1}})+\rho_{2}F\left(T^{k+1}-\frac{m_{2}}{\rho_{2}}\right)\;\;\;\;\;\;\;+\rho_{3}F\left({d_{x}}^{T}\right).*F(d_{x}V_{n}-U^{k+1}+\frac{m_{3}}{\rho_{3}})$$

By left division of the matrix, we obtain the following:(29)$$F\left(N^{k+1}\right)=\left[\rho_{1}F\left({d_{L}}^{T}\right).*F\left(d_{L}V_{n}-H^{k+1}+\frac{m_{1}}{\rho_{1}}\right)+\rho_{2}F\left(R^{k+1}-\frac{m_{2}}{\rho_{2}}\right)\;\;\;+\rho_{3}F\left({d_{x}}^{T}\right).*F\left(d_{x}V_{n}-M^{k+1}+\frac{m_{3}}{\rho_{3}}\right)\right].\;\;\;\;\;\;\;/\left[\rho_{1}F\left({d_{L}}^{T}\right).*F\left(d_{L}\right)+\rho_{2}+\rho_{3}F\left({d_{x}}^{T}\right).*F\left(d_{x}\right)\right]$$

Then, the inverse Fourier transform of Formula (29) was implemented to obtain the expression for stripe noises N:(30)$$N^{k+1}=F^{-1}\{\left[\rho_{1}F\left({d_{L}}^{T}\right).*F\left(d_{L}V_{n}-H^{k+1}+\frac{m_{1}}{\rho_{1}}\right)+\rho_{2}F\left(R^{k+1}-\frac{m_{2}}{\rho_{2}}\right)+\rho_{3}F\left({d_{x}}^{T}\right).*\;F\left(d_{x}V_{n}-M^{k+1}+\frac{m_{3}}{\rho_{3}}\right)\right]./\left[\rho_{1}F\left({d_{L}}^{T}\right).*F\left(d_{L}\right)+\rho_{2}+\rho_{3}F\left({d_{x}}^{T}\right).*F\left(d_{x}\right)\right]\}$$
where .* is the point multiplication of two matrices; “./” is the point division of two matrices; “F()” is the Fourier transform; “$F^{-1}()$” is the inverse Fourier transform. Note that the complete convolution of a matrix will change its size. It is highly necessary to conduct normalization during computing. After each iteration, the Lagrange multipliers must be updated by the following [33]:(31)$$\left\{\begin{array}{c} m_{1}^{k+1}=m_{1}^{k}+\rho_{1}\left(d_{L}V_{n}{-d}_{L}N^{k+1}-H^{k+1}\right)\\ m_{2}^{k+1}=m_{2}^{k}+\rho_{2}\left(N^{k+1}-R^{k+1}\right)\\ m_{3}^{k+1}=m_{3}^{k}+\rho_{3}\left(d_{x}V_{n}{-d}_{x}N^{k+1}-M^{k+1}\right) \end{array}\right.$$

Finally, the noise component $N^{k+1}$ in the original image is obtained, and the final low-rank part image $V_{n}\prime$ with the striping noise removed is obtained by subtracting $N^{k+1}$ from $V_{n}$.

The following equation can ultimately obtain the denoised image matrix of the original input image:(32)$$I_{C}={DS}_{n}+V_{n}\prime$$
where $I_{C}$ Is the denoised image. As the denoising algorithm was only applied to the final low-rank part image $V_{n}$. In this chapter, the detailed texture information preserved in ${DS}_{n}$ remains unaffected by the denoising algorithm.

## 4. Experimental Results

The proposed method was compared with four state-of-the-art methods on three different image datasets in experimental comparisons. The contrastive methods were as follows: multi-scale guided filter (MSGF) [34] for stripe noise, wavelet transform coupled with gradient equalization (WAGE) [8], Effective Strip Noise Removal for Low-Textured Infrared Images Based on 1D Guided Filtering (1D-GF) [35], and FPN-Based Learning Convolutional Network (FLCN) [36]. To further demonstrate the effectiveness of the proposed method, we conducted ablation experiments in the comparative experiment. Specifically, we performed denoising experiments using an optimization model on images subjected to one round of LatLRR decomposition without noise extraction, referred to as the Non-MIDILATLRR method.

All the image data were shot using a LUSTER TB640-CL refrigerated medium wavefront infrared camera. All experiments were run in M.A.T.L.A.B. (R2020b) on a computer with an Intel i7 9750H six-core twelve-thread processor @4.5 GHz and 16 GB of RAM.

We focused on evaluating the denoising effectiveness of the experimental data in terms of edge details and overall noise reduction, using both subjective and objective methods. Due to a lack of available real images for reference, we selected no-reference evaluation metrics, including noise reduction (NR) [37,38], mean relative deviation (MRD) [37,39], and image distortion (I.D.) [40,41]. The NR metric, defined by Equation (33), reflects the overall performance of the denoised image. The MRD metric, defined by Equation (34), reflects the ability to preserve image information in non-stripe regions. The ID metric, as defined by Equation (35), indicates the level of distortion present in the resulting denoised image. The effectiveness of denoising is positively correlated with the NR and ID metrics and negatively correlated with the MRD metric. Therefore, these metrics enable accurate evaluation of the denoising performance without the need for reference images.
(33)$$\left\{\begin{array}{c} NR=N_{0}/N_{1}\\ N=\sum_{i=0}^{k}meanP\left(u_{i}\right) \end{array}\right.$$
where $N_{0}$ and $N_{1}$ stand for the value of N in the original and de-striped images, respectively. ui is the frequency component produced by stripes. N corresponds to the total power of the noise produced by stripes in the mean power spectrum.
(34)$$MRD=\frac{1}{MN}\sum_{i=1}^{MN}\frac{\left|z_{i}-g_{i}\right|}{g_{i}}\times 100$$
where $g_{i}$ and $z_{i}$ are the pixel values of point i in the original image and the image after stripe noise removal, respectively. Additionally, MN represents the number of all pixels in the selected area.
(35)$$\left\{\begin{array}{c} ID=S_{1}/S_{0}\\ S=\sum_{j\neq 1}^{N-1}meanP\left(u_{i}\right) \end{array}\right.$$
where $S_{0}$ and $S_{1}$ stand for the value of S in the original image and the de-striped image, respectively. $u_{i}$ stands for the raw image caused by the frequency component without stripes. S stands for the total power of the clean image in the mean power spectrum.

### 4.1. Parameter Analysis

Using image x as an example, a sensitivity analysis was conducted on the coefficients of the three regular terms in the multi-sparse constraint representation model to verify the importance of the key parameters for the proposed method. We selected the peak signal-to-noise ratio (PSNR) as the representative full-reference evaluation index for this experiment to evaluate the effectiveness of parameter selection. For MIDILatLRR decomposition, we chose four times based on the adaptive decomposition level to balance the speed of operation and the denoising effect. The sparse noise item parameter λ in LatLRR decomposition was set to 1, and the λ1 parameter in the multi-sparse constraint representation model was also set to 1 based on experimental experience. The relationship between PSNR and the regular terms λ2 and λ3 is shown in Figure 10 [42]. The results in Figure 10 demonstrate that the selected λ2 and λ3 significantly impacted the denoising performance. According to the experimental results, λ1 was determined to be 1, while λ2 and λ3 should be set to 0.8 and 1.2, respectively. After experimentation, the penalty coefficient was set as ρ1 = ρ2 = ρ3 = 0.15.

### 4.2. Experimental Contents

Four different infrared images were selected as experimental objects to demonstrate the universality and effectiveness of the proposed method. The first image, as shown in Figure 11a, contains a human figure with a significant grayscale difference from the background and an object with vertical edge features with a small grayscale difference from the background. In the second image, depicted in Figure 11b, the main structure of a wall in a building is included, in which the texture details have a small grayscale difference when compared to the stripe noise, and there are numerous small vertical detailed textures in the windows and wall structures. The third image, illustrated in Figure 11c, contains a vehicle with relatively smooth information and fewer vertical texture features. Lastly, the fourth image, displayed in Figure 11d, depicts a complex scene comprising human figures, vehicles, trees, and the ground, in which the texture information is intricate and subtle objects with vertical texture features exist. All four images contain various stripe noises with different grayscale and intensity, and large amounts of effective texture information have low contrast with the stripe noise. Resultantly, it is sufficiently demonstrated that the proposed method can attain excellent results.

#### 4.2.1. Ablation Experiments

In this section, the ablation experiment of the proposed method is presented. The results from the Non-MIDILatLRR method are shown, where only the first-level LatLRR decomposition is used, and the MSCR algorithm is directly utilized for subsequent denoising. As shown in Figure 12, the Non-MIDILatLRR and proposed method exhibit good denoising performance. However, Figure 10a highlights that the information on vertical edge features in the image is relatively blurred with low contrast, and there is a loss of information of micro-texture details (enclosed by the red line), where the MIDILatLRR method shows superiority in preserving image details and texture.

As demonstrated in Figure 13a,b, good denoising performance is exhibited by both the Non-MIDILATLRR method and the proposed method. Nonetheless, significant blurring of texture details on building walls (enclosed by the red line), as well as the substantial loss of micro-texture details, particularly those with vertical edge features such as doors and windows, is revealed by Figure 13a for the Non-MIDILATLRR method. In contrast, the proposed method performs excellent denoising and preserves the details and texture of buildings in the image.

As demonstrated in Figure 14, excellent performance in removing stripe noise is exhibited by both the Non-MIDILatLRR method and the proposed method. However, it can be observed in Figure that the texture of the tree (enclosed by the red line) in the image’s background becomes more blurred after denoising with the Non-MIDILatLRR method. The complex texture is lost, and a substantial loss of edge information can be observed, particularly for effective information with vertical edges (enclosed by the red line), which also suffers from some contrast loss.

As depicted in Figure 15, the stripe noise in the original image has been effectively removed by both the Non-MIDILatLRR method and the proposed method. Nevertheless, it can be observed in Figure 15 that there is a severe loss of effective information with the vertical edge features (enclosed by the red line) when using the Non-MIDILatLRR method, and a limited preservation of details, such as the folds on the person’s clothing, is observed. In contrast, the proposed method preserves detailed information with vertical texture features and effective information with smaller edge texture features.

#### 4.2.2. Comparison Experiments

To validate the effectiveness of the proposed method, a series of experiments were conducted on images that contained stripe noise. Figure 16 shows the results of the first experiment. As demonstrated in Figure 16a, the denoising effect of MSGF was relatively poor as it could not effectively identify and remove irregular stripe noise with small horizontal gradient changes. Similarly, as shown in Figure 16b, the WAGE method retained a small amount of stripe noise, resulting in incomplete denoising. Furthermore, Figure 16c illustrates that FLCN caused over-smoothing of the original image in the horizontal direction, leading to relatively blurry edges of the person and vertical features of objects (enclosed by the red line).

On the other hand, Figure 16d presents the 1D-GF method, which also had a good denoising effect. However, two white lines (enclosed by the red line) appeared above the person. Additionally, there was a slight loss of detailed texture in some areas of the image and a larger loss of information in certain vertical texture features (enclosed by the red line). In contrast, Figure 16e shows that the proposed method can effectively remove various types of stripe noise while preserving edge texture information as much as possible. This comparison fully demonstrates the superiority of the proposed method in removing stripe noise and validates its effectiveness.

The second experiment is presented in Figure 17, which depicts an original image containing the main structural components of a building with numerous vertical structures and nearly vertical effective texture information. As shown in Figure 17a and b, despite preserving the edges and details of the image to some extent, the MSGF and WAGE algorithms exhibited poor denoising performance, leaving partial stripe noise in the image. Figure 17c illustrates that excessive smoothing by FLCN caused some detailed texture information to be lost, including the disappearance of vertical edge features, such as the windows of the building, during the denoising process.

Compared to the previous methods, Figure 17d demonstrates that the 1D-GF method preserved the details of the building to a certain extent. However, there were still significant losses in some vertical edge feature information, such as the gaps between windows, as enclosed by the red line. As demonstrated in Figure 17e, the proposed method not only effectively removed stripe noise but also achieved outstanding performance in preserving the edges and details of the building. These results further support the superiority of the proposed method.

The third experiment is presented in Figure 18, which includes the side view of a car and background elements, such as trees and pillars. As illustrated in Figure 18a,b, poor denoising performance was exhibited by MSGF and WAGE among all the methods tested. Although they preserved the edges and details of the image to some extent, the noise was not entirely eliminated. Figure 18c shows that although FLCN performed well in denoising, it significantly attenuated the edge features, as evidenced by the visible blurring of the tree branches circled in Figure (enclosed by the red line).

Although the 1D-GF method had decent denoising performance, there were still certain losses in the detailed features with vertical edge characteristics, as clearly seen in the pillar region of the image. Moreover, blocky grayscale losses were observed at the intersection between the pillars and the car, as enclosed by the red line. As demonstrated in Figure 18e, the proposed method not only preserved the edge information but also effectively removed stripe noise. These results provide further evidence of the superiority of our method.

The final experiment, as illustrated in Figure 19, consists of elements such as people, vehicles, trees, and the ground. It is observed from Figure 19a,b that the other methods outshined MSGF and WAGE, as they failed to remove some stripe noise. As highlighted in red, Figure 19c indicates that FLCN induced blurring of the detailed edge parts of the denoised image to a certain extent (enclosed by the red line). Meanwhile, Figure 19d reveals that the 1D-GF method performed better than the previous methods in denoising and preserving details. However, it still fell short of retaining information with vertical textures, resulting in blocky blurriness or information loss enclosed by a red line. The proposed method, presented in Figure 19e, demonstrated outstanding performance in denoising and preserving details with vertical edge features. These results fully reflect the superiority of our method, which has promising potential for practical applications.

Table 1 compares the NR, MRD, and ID values of the images denoised by the four methods. The optimal value of each metric is shown in bold.

Table 1 presents the results for different denoising methods regarding NR, MRD, and ID. The proposed method achieved the best results regarding NR and MRD, indicating its effectiveness in denoising images with stripe noise. Our method also showed good performance in ID, which suggests its ability to preserve useful information from the original image while removing noise. In contrast, some methods produced higher ID values than ours but failed to eliminate noise completely. The subjective evaluation of Figure 16, Figure 17, Figure 18 and Figure 19 shows that the proposed method effectively removes stripe noise without introducing severe distortion. Moreover, it preserves edge details with an ID value close to 1 and distinguishes between useful information and noise with vertical edge textures. These results demonstrate that the proposed method effectively removes stripe noise while retaining edge details and preserving the maximum amount of information in the image.

## 5. Conclusions

This paper verifies that LatLRR is applicable for separating stripe noise and proposes a multi-level image decomposition method, MIDILatLRR. Our method fully utilizes the global low-rank properties of stripe noise and adaptively decomposes noisy images into multiple levels to separate noise from useful information. By concentrating stripe noise in the final low-rank part image through MIDILatLRR, denoising the last low-rank partial image is required. In contrast, salient part images are retained by preserving the extracted edge details.

The differences in smoothness and directionality between stripe noise and effective information in the last-level low-rank partial image are used simultaneously to construct sparse terms of the MSCR model by adopting the L1-norm for the sparsity constraint. To solve the proposed optimization model, the ADMM algorithm is introduced. Extensive experiments demonstrate the superiority of our method. However, our method still has some shortcomings, such as its relatively slow computational speed. In the future, we will concentrate on resolving these concerns.

## Figures and Tables

**Figure 1 sensors-23-06786-f001:**
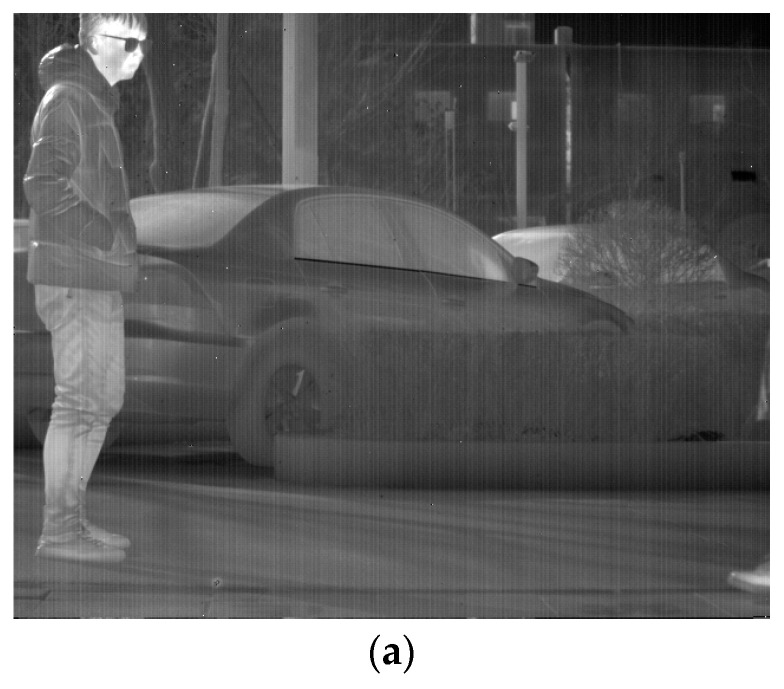
The various view of LatLRR decomposition for noisy images. (**a**) The original noisy input image. (**b**) Low-rank component. (**c**) Salient component. (**d**) Sparse noise. (**e**) The proportion of L1 norm of each column of sparse noise component to the original image.

**Figure 2 sensors-23-06786-f002:**
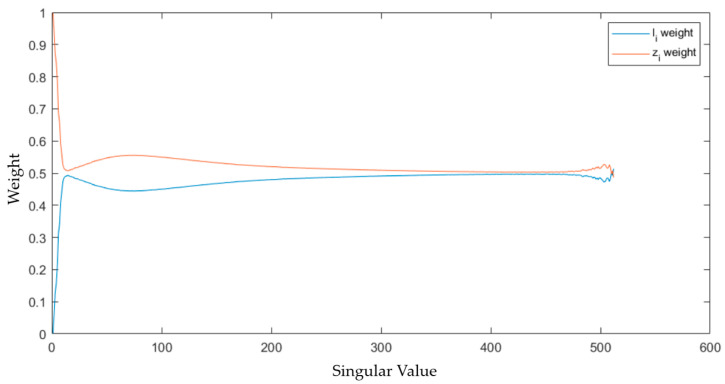
The proportion of the singular value weighted of salient and low-rank components to the original image matrix.

**Figure 3 sensors-23-06786-f003:**
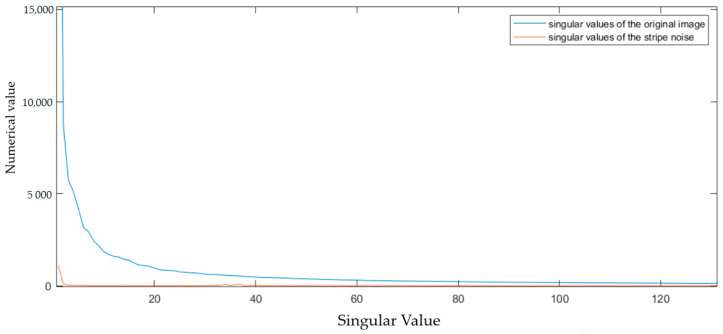
Striped noise singular values compared to original image singular values.

**Figure 4 sensors-23-06786-f004:**
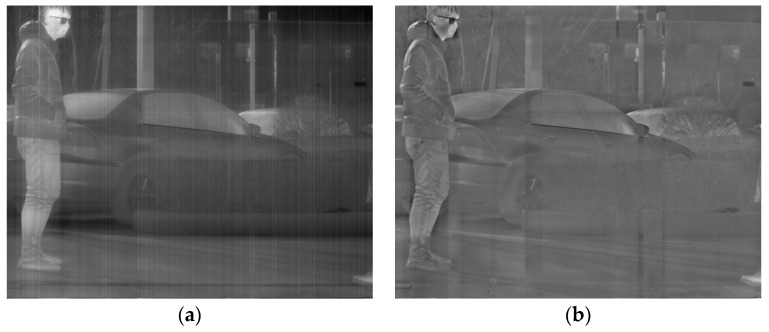
(**a**) Low-rank component. (**b**) Normalization of salient component.

**Figure 5 sensors-23-06786-f005:**
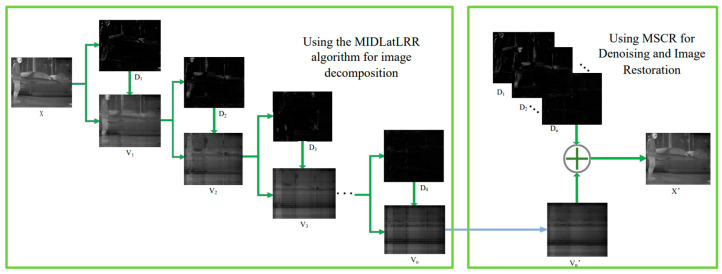
Overall flowchart of the algorithm in this paper.

**Figure 6 sensors-23-06786-f006:**
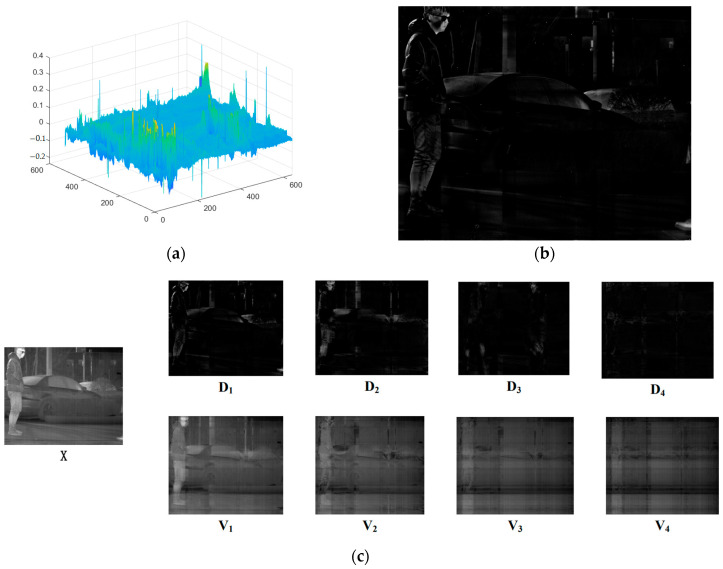
(**a**) The original salient component matrix. (**b**) The image after extracting positive elements without distortion mapping. (**c**) MIDILATLRR multi-level image decomposition results.

**Figure 7 sensors-23-06786-f007:**
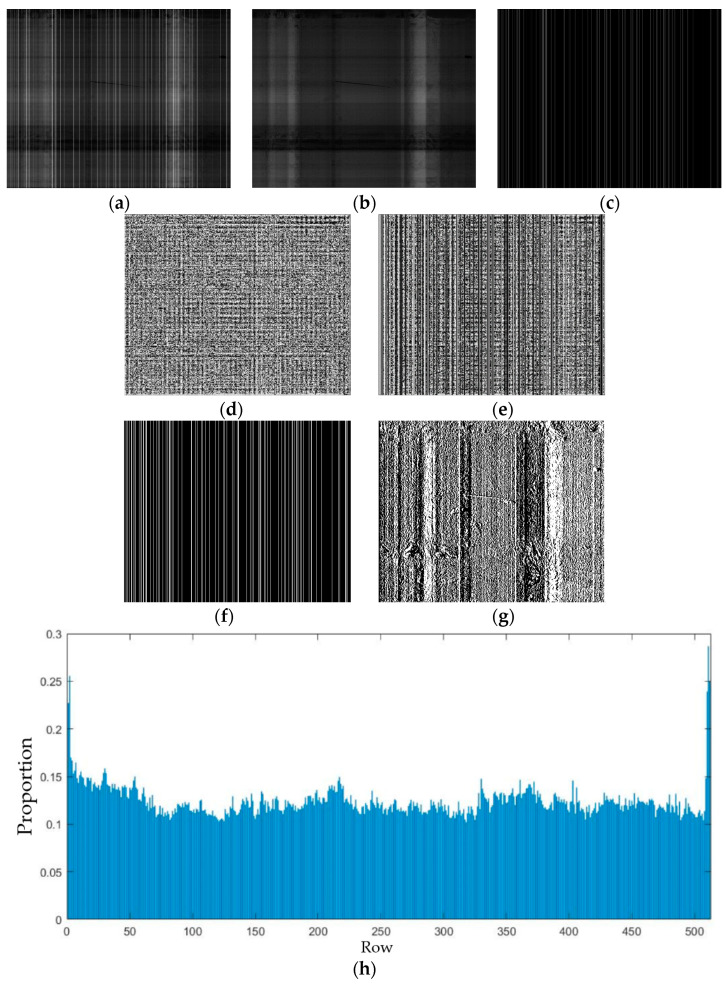
Difference between effective smooth information and stripe noises: (**a**) Original final low-rank part image. (**b**) Effective information components. (**c**) Noise components. (**d**) Edge sharpening of effective information. (**e**) Edge sharpening of the original image. (**f**) Horizontal gradient of noise component. (**g**) Horizontal gradient of effective information. (**h**) Proportion of the L1-norm for each row of information components in the L1-norm for the edge gradient of the original image. (**i**) Proportion of the L1-norm for each row of information components in the L1-norm for the horizontal gradient of the original image. (**j**) Proportion of the L1-norm for each column of the original image.

**Figure 8 sensors-23-06786-f008:**
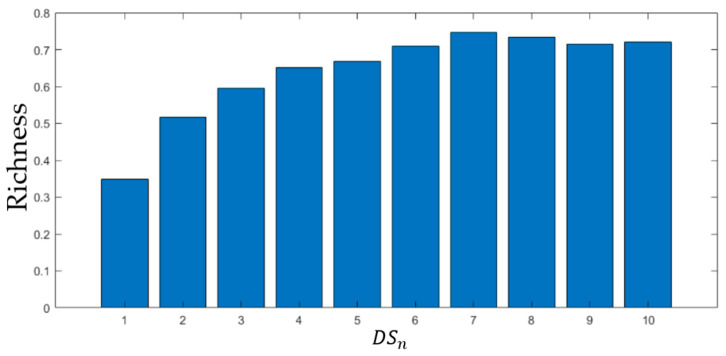
Illustration of the richness of ${DS}_{n}$.

**Figure 9 sensors-23-06786-f009:**
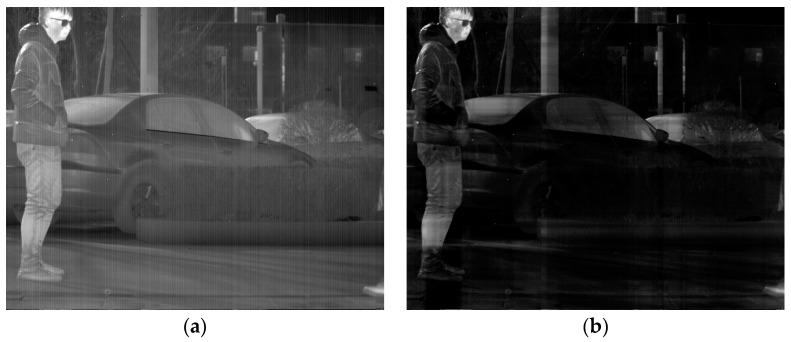
Adaptive decomposition result image: (**a**) Original image; (**b**) ${DS}_{n}$ Extracted by MIDILatLRR.

**Figure 10 sensors-23-06786-f010:**
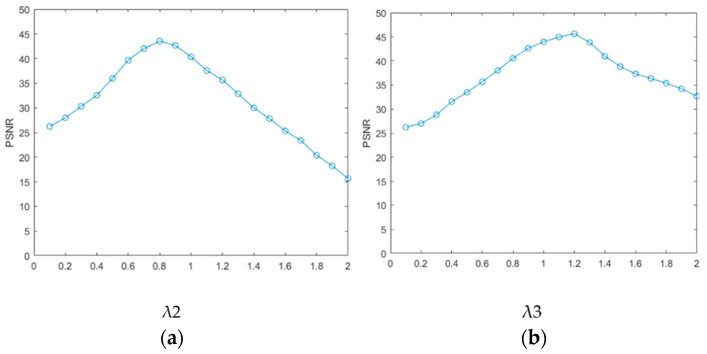
Influence of regular terms on the PSNR: (**a**) relationship between λ2 and PSNR; (**b**) relationship between λ3 and PSNR.

**Figure 11 sensors-23-06786-f011:**
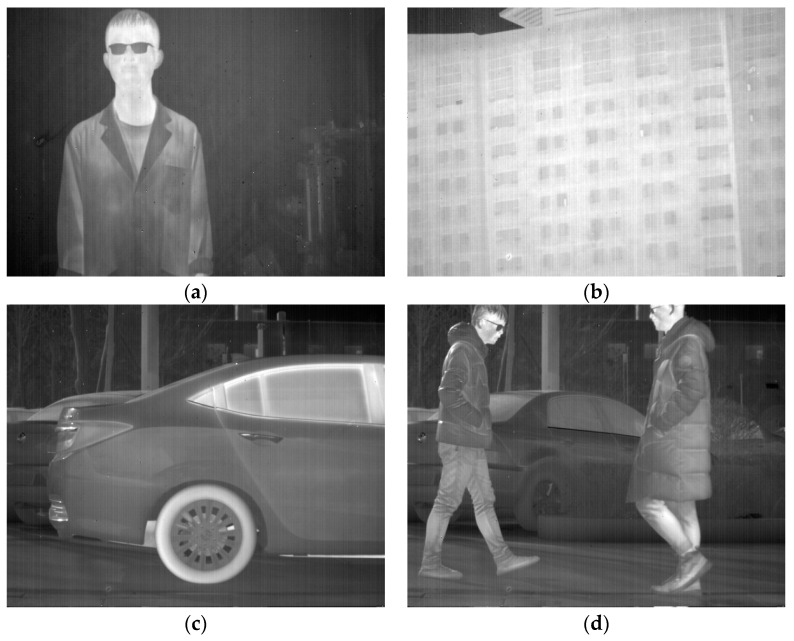
Experimental images: (**a**) A person. (**b**) A single building. (**c**) A car. (**d**) A complex scene image.

**Figure 12 sensors-23-06786-f012:**
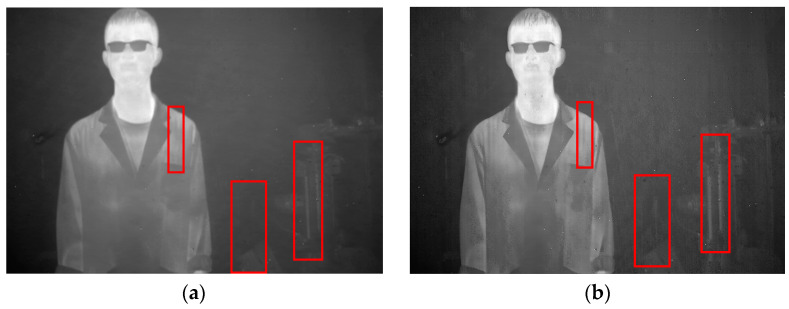
Denoising effects of ablation experiment on an image of a person: (**a**) Non-MIDILatLRR; (**b**) proposed method.

**Figure 13 sensors-23-06786-f013:**
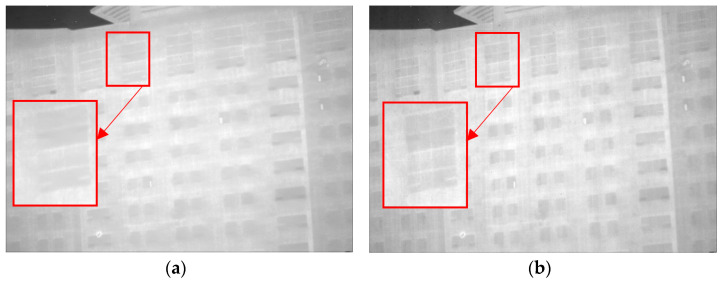
Denoising effects of ablation experiment on an image of a single building: (**a**) Non-MIDILatLRR; (**b**) proposed method.

**Figure 14 sensors-23-06786-f014:**
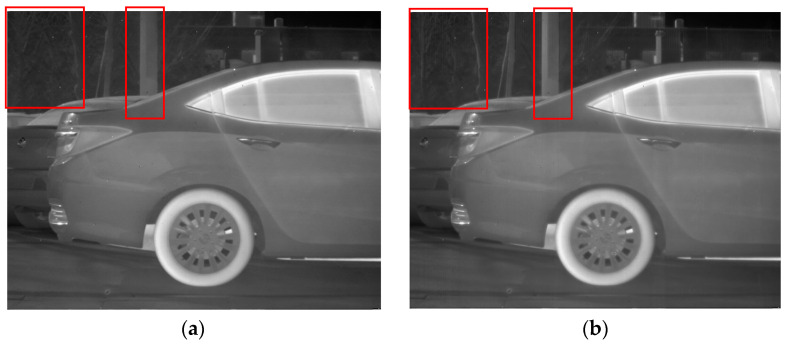
Denoising effects of ablation experiment on an image of a car: (**a**) Non-MIDILatLRR; (**b**) proposed method.

**Figure 15 sensors-23-06786-f015:**
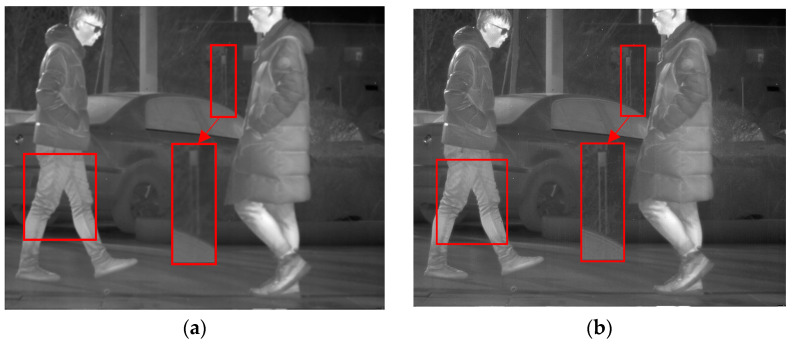
Denoising effects of ablation experiment on an image of a complex scene image: (**a**) Non-MIDILatLRR; (**b**) proposed method.

**Figure 16 sensors-23-06786-f016:**
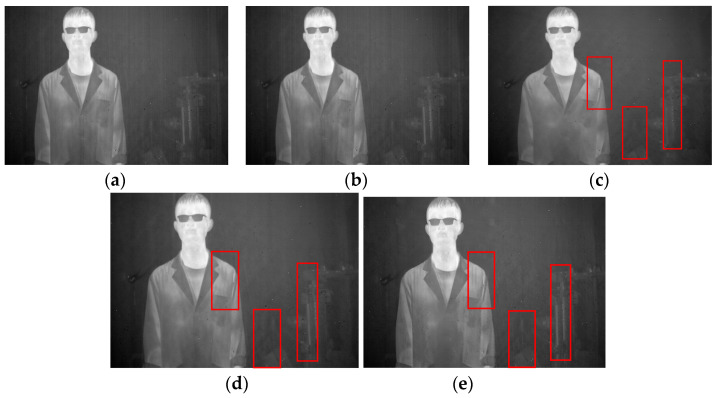
Denoising effects of different methods on an image of a person: (**a**) MSGF; (**b**) WAGE; (**c**) FLCN; (**d**) 1D-GF; (**e**) proposed method.

**Figure 17 sensors-23-06786-f017:**
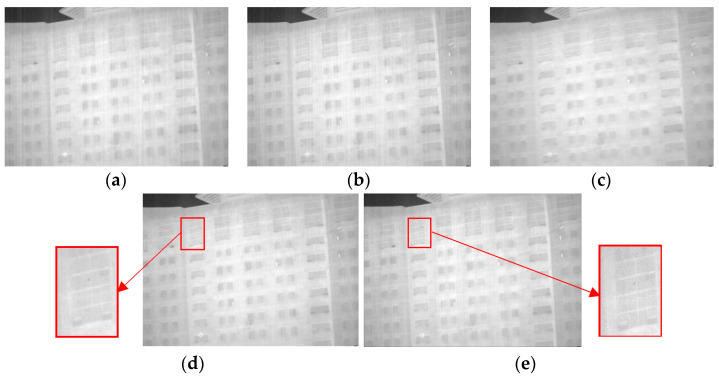
Denoising effects of different methods on an image of a single building: (**a**) MSGF; (**b**) WAGE; (**c**) FLCN; (**d**) 1D-GF; (**e**) proposed method.

**Figure 18 sensors-23-06786-f018:**
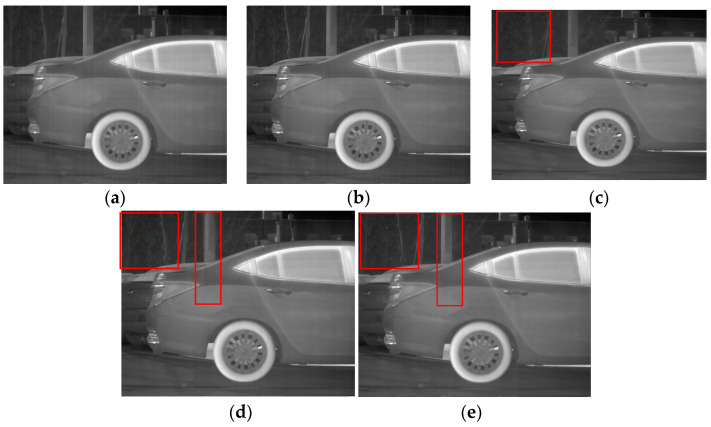
Denoising effects of different methods on an image of a car: (**a**) MSGF. (**b**) WAGE; (**c**) FLCN; (**d**) 1D-GF; (**e**) proposed method.

**Figure 19 sensors-23-06786-f019:**
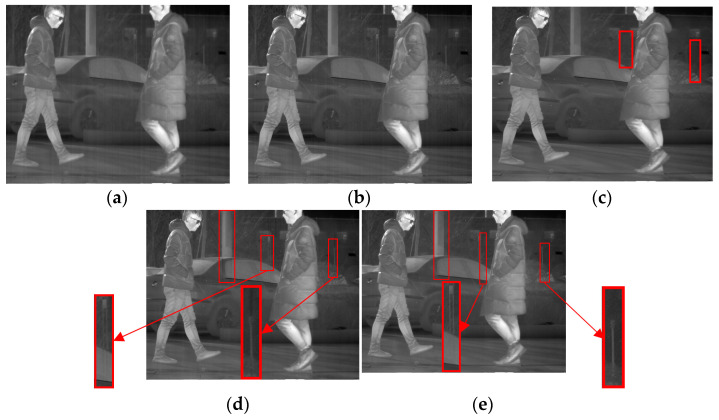
Denoising effects of different methods on an image of a complex scene image: (**a**) MSGF. (**b**) WAGE; (**c**) FLCN; (**d**) 1D-GF; (**e**) proposed method.

**Table 1 sensors-23-06786-t001:** Metrics of different methods on different images.

Image	Indices	MSGF	WAGE	FLCN	1D-GF	Non-MIDILATLRR	Proposed Method
Person	NR	2.03	2.49	3.72	4.05	4.02	**4.08**
	MRD (%)	**2.92**	3.68	3.95	4.12	4.12	3.09
	ID	0.999	0.995	0.978	0.972	0.984	0.988
Building	NR	2.43	2.75	3.67	3.86	3.89	**3.98**
	MRD (%)	**3.94**	4.33	4.87	4.30	4.28	4.13
	ID	0.999	0.992	0.978	0.979	0.983	0.986
Car	NR	3.29	3.35	3.42	3.49	3.47	**3.52**
	MRD (%)	2.76	2.57	3.40	2.81	2.86	**2.47**
	ID	0.999	0.992	0.976	0.986	0.985	0.991
Complex scene image	NR	3.08	3.16	3.43	3.39	3.38	**3.43**
	MRD (%)	3.18	2.75	4.12	2.62	2.59	**2.28**
	ID	0.999	0.992	0.981	0.991	0.989	0.994

## Data Availability

Due to the nature of this research, participants of this study did not agree for their data to be shared publicly, so supporting data are not available.

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
