# Peer review of "An Innovative Approach for Removing Stripe Noise in Infrared Images"

_sensors, 2023, doi:10.3390/s23156786_

Round 1

Reviewer 1 Report

In this work, the author propose a new method to improve the image with stripe noise. The image is significantly improve compared with traditional method. This paper is a good  reference for the researchers in this area. It seens acceptable to publish in Sensors. I have a small question on the technical detail. In page 22  the comparasion on different methods is shown.  Can the author give a criteria of their method quantitatively? That means how to choose a certain method for a cetain situation.

Reviewer 2 Report

The paper is of interest. However, some changes are needed before the next round of revision. In particular, I’m referring to:

^ The Introduction section is missing the discussion of an important paper recently published, which fits the core of the paper under review, i.e.: DOI 10.1080/17686733.2020.1786640

^ At the end of the “Introduction” section, please explain briefly how the paper will be developed in the subsequent sections.

^ The paper needs to be checked accurately from the English point of view. See, e.g., line 154 on page 4: “…if a infrared image…” -> It should be “an image.” However, this is only an example of many reported in the paper!

^ Fig. 1b, c, d: What do the X, Y, and Z axes represent? The same can be said about Fig. 6a.

^ The infrared images shown in the paper, in which spectral range of the “infrared” were collected?

^ X-axis of Fig. 10a, b should be λ2 and λ3.

^ A couple of red rectangles above page 20 should be deleted and positioned correctly.

^ What happens when a surface has a very low emissivity value? For example, could you try an aluminum surface as a background of polished metallic objects (with complex 3D surfaces)?

The paper needs to be checked accurately from the English point of view. See, e.g., line 154 on page 4: “…if a infrared image…” -> It should be “an image.” However, this is only an example of many reported in the paper!
